# Double Blast Wave Primary Effect on Synaptic, Glymphatic, Myelin, Neuronal and Neurovascular Markers

**DOI:** 10.3390/brainsci13020286

**Published:** 2023-02-08

**Authors:** Diego Iacono, Erin K. Murphy, Cheryl D. Stimpson, Fabio Leonessa, Daniel P. Perl

**Affiliations:** 1DoD/USU Brain Tissue Repository and Neuropathology Program, Uniformed Services University (USU), Bethesda, MD 20814, USA; 2Department of Neurology, F. Edward Hébert School of Medicine, Uniformed Services University (USU), Bethesda, MD 20814, USA; 3Department of Pathology, F. Edward Hébert School of Medicine, Uniformed Services University (USU), Bethesda, MD 20814, USA; 4Neuroscience Graduate Program, Department of Anatomy, Physiology, and Genetics, F. Edward Hébert School of Medicine, Uniformed Services University (USU), Bethesda, MD 20814, USA; 5Henry M. Jackson Foundation for the Advancement of Military Medicine, Inc., Bethesda, MD 20814, USA; 6Neurodegenerative Clinics, National Institute of Neurological Disorders and Stroke (NINDS), NIH, Bethesda, MD 20814, USA

**Keywords:** blast-wave primary effect, traumatic brain injury, brain region-based blast-sensitivity, molecular changes, brainstem, subclinical status, neurodegeneration

## Abstract

Explosive blasts are associated with neurological consequences as a result of blast waves impact on the brain. Yet, the neuropathologic and molecular consequences due to blast waves vs. blunt-TBI are not fully understood. An explosive-driven blast-generating system was used to reproduce blast wave exposure and examine pathological and molecular changes generated by primary wave effects of blast exposure. We assessed if pre- and post-synaptic (synaptophysin, PSD-95, spinophilin, GAP-43), neuronal (NF-L), glymphatic (LYVE1, podoplanin), myelin (MBP), neurovascular (AQP4, S100*β*, PDGF) and genomic (DNA polymerase-*β*, RNA polymerase II) markers could be altered across different brain regions of double blast vs. sham animals. Twelve male rats exposed to two consecutive blasts were compared to 12 control/sham rats. Western blot, ELISA, and immunofluorescence analyses were performed across the frontal cortex, hippocampus, cerebellum, and brainstem. The results showed altered levels of AQP4, S100*β*, DNA-polymerase-*β*, PDGF, synaptophysin and PSD-95 in double blast vs. sham animals in most of the examined regions. These data indicate that blast-generated changes are preferentially associated with neurovascular, glymphatic, and DNA repair markers, especially in the brainstem. Moreover, these changes were not accompanied by behavioral changes and corroborate the hypothesis for which an asymptomatic altered status is caused by repeated blast exposures.

## 1. Introduction

Exposure to single or multiple explosive-driven blasts are life-threating events that may occur to deployed military personnel and civilian populations in war and non-war zones. In particular, the last 20 years of war in Afghanistan and Iraq territories (Operation Enduring Freedom [OEF], Operation Iraqi Freedom [OIF] and Operation New Dawn [OND]) have been characterized by the terroristic use of blast-generating weapons, and in particular by the use of improvised explosive devises (IEDs) by the enemy. Blast exposures generated by IED detonations have been indeed among the most recurrent battlefield events used by the enemy during the OEF/OIF/OND periods [1,2,3]. Sadly, it has been estimated that around 7000 deaths among 2.7 million deployed Service Members over 20 years have been caused by IEDs events.

In medical terms, multiple blast exposures may be associated with a wide spectrum of clinical manifestations, especially persistent neurological and psychiatric phenomena [4,5,6,7]. These neurological and psychiatric chronic phenomena have been hypothesized to be caused by the possible cumulative effects of multiple subconcussive, concussive, and blunt traumatic brain injury (TBI) events, which often and simultaneously, occur during these types of explosions [8,9]. In particular, IED blast wave exposures have been hypothesized to trigger neuropathological and long-term brain molecular consequences capable to induce various functional and neurochemical anomalies and culminate later into manifestations of a wide spectrum of neurological and psychiatric illnesses, including neurodegenerative disorders [10,11,12]. In particular, long-term blast-related neuropsychiatric disorders reported in war veterans (https://www.cccneb.edu/veterans on 6 February 2023) are phenomenologically complex and pathophysiologically puzzling [13,14]. In fact, blast-related brain disorders can range from clinically isolated disorders such as anxiety disorder, hyperirritability, vestibular impairment, etc. to very complex and hybrid clusters of different clinical phenomena including, among others, chronic debilitating headache and migraine syndromes, severe balance and motor disorders, major cognitive deficits, untreatable sleep disorders, emotional disarrays, pervasive changes of personality, socially altered behavior, prolonged post-traumatic stress disorders (PTSD), suicidal ideation, suicide attempts, and suicide [15,16,17,18].

More specifically, blast-TBI exposure may affect the central nervous system (CNS), and the brain in particular, through different pathomechanisms associated mainly, although not exclusively, with sudden acceleration/deceleration forces impacting the head/skull, shrapnel penetrations, abrupt over-threshold changes of the intracranial pressure and brief or prolonged loss of consciousness (LOC) [19,20]. Nonetheless, the specific pathogenetic consequences on the brain and the CNS in general as distinctively due to the primary effect of blast waves [21] are not completely understood [22,23,24,25]. Here, it is important to remind that the elevated frequency of multiple blast exposures reported by active duty service members and war veterans, is of special relevance not only in terms of serious clinical sequelae among active military members [26], but also for the possible societal and health system concerns due to the higher frequency of persistent symptomatology among war veterans (https://www.moaa.org/content/publications-and-media/news-articles/2019-news-articles/pentagon-ordered-to-add-blast-exposure-to-troops-medical-histories on 6 February 2023).

Preclinical and experimental studies in rodents and larger mammals (i.e., swine) have demonstrated a variety of short- and long-term brain consequences due to single and repeated blasts [27,28,29]. A wide range of different devices and experimental paradigms have been used to model blast wave forces as possibly experienced on the battlefield. Most of these devices have been based on the gas-generated pressure wave tube methods [30]. Some labs, though, have used a much more battlefield-like blast wave-generating method to better and specifically examine the consequences of the blast wave primary effects exposing the animals to explosive-driven blast waves in an open-field setting, which is indeed a more realistic way to replicate real-life blast events [31,32,33,34,35,36,37,38,39,40,41,42]. It is important to notice that, while these explosive-driven methods represent a more realistic approach in terms of blast wave generation and possible related effects, they are also much more difficult to carry out since large amounts of explosives, access to appropriate testing facilities, and specialized personnel are required [37]. This is actually one of the reasons why most of the blast-TBI investigators opted for the more easily accessible and economically affordable pressure-wave systems to study the blast consequences on the brain. However, while these pressure-wave systems have been undoubtedly valuable tools of blast research, they unfortunately present some intrinsic limitations, which have reduced their applicability and comparability to human studies [43].

Recently, our lab reported a series of new findings obtained from an experimental setting that employed an explosive-driven repeated blast paradigm and scenario [40]. The employment of this more battlefield-like experimental paradigm scenario has been a very useful experimental method to more precisely investigate the direct neuropathological and neuromolecular consequences generated by the primary effects of explosive-driven double blast wave exposure. In addition to the advantage of using an explosive-driven blast wave-generating system, our previous analyses applied a more detailed neuroanatomically based strategy aiming to identify a differential blast-sensitivity (or blast-reactivity) possibly present across the different regions of the mammalian brains. By measuring the expression levels of proteins ordinarily present in different regions of wild type rat brains (i.e., phosphorylated-Tau [pTau], amyloid precursor protein [APP], glial fibrillary acidic protein [GFAP]) and other proteins known to be involved in different neurodegenerative conditions, we observed significant differences, especially in terms of pTau levels, across the different brain regions examined (frontal cortex [FCtx], hippocampus [H], cerebellum [CRB], brainstem [BS]). Specifically, increased levels of pTau in the H, CRB, and BS, together with increased levels of APP in the CRB, of double-blast vs. sham animals were found. Remarkably, these brain region-based molecular changes in double-blast vs. sham animals measured after 15 days from the second consecutive blast were not associated with significant behavioral, cognitive, or neurological abnormalities, nor with obvious neurohistopathological lesions as detectable by light microscopy.

Based on those previous findings, we hypothesized that the molecular changes found in double blast vs. sham animals characterize, as a whole, an asymptomatic subclinical neuromolecular condition, which we termed asymptomatic blast-induced molecular altered status (ABIMAS), whose persistence or exacerbation due to further detrimental events such as additional blast- or blunt-TBI events, genetic and biological risk factors, or other environmental stressors, could culminate later into a higher risk for the manifestation of various neurological and psychiatric disorders [44,45]. Indeed, the ABIMAS condition seems to fit well with those human case reports often describing delayed and unusual neurological and psychiatric manifestations among military personnel with a relatively remote history of multiple blast-TBI exposure [46]. Actually, late clinical manifestations related to blast-exposure have been described in civilian populations as well [47,48,49].

We extended our previous analyses by measuring the expression levels of other brain-related molecules in double-blast (2×B) vs. sham/control (Ctl) rats to further characterize the inherent differential blast sensitivity across different regions of the brain and support the main hypothesis for which an altered subclinical neuromolecular status is present as a direct molecular outcome generated by repeated primary blast wave exposure.

Based on their well-known involvement in various cognitive, behavioral, and neurodegenerative mechanisms, we measured the expression levels of some synaptic, glymphatic, myelin, neuronal, neurovascular and DNA-repair markers across four different regions of the brain (FCtx, H, CRB and BS) to further support our hypothesis. In particular, we measured the soluble phase expression levels of the following proteins grouped by neurobiological function:

### 1.1. Synaptic Markers

Synaptophysin (SYN), an integral membrane glycoprotein expressed in presynaptic vesicles of neurons and adrenal medulla [50];

Postsynaptic density protein 95 (PDS-95), a post-synaptic protein member of the membrane-associated guanylate kinase (MAGUK) family encoded by the DLG4 (discs large homolog 4) gene. PDS-95 is involved in anchoring synaptic proteins and plays an important role in synaptic plasticity and the stabilization of synaptic changes during long-term potentiation [51];

Spinophilin (SPIN), a dendritic spine protein, a regulatory subunit of phosphatase-1 catalytic subunit highly enriched in dendritic spines that receive most of the excitatory input in the central nervous system [52];

Growth associated protein 43 (GAP-43), an axonal and presynaptic terminal protein expressed at high levels in neuronal growth cones during development, and axonal regeneration, and is phosphorylated after long-term potentiation and after learning [53].

### 1.2. Glymphatic Markers

Lymphatic vessel endothelial receptor-1 (LYVE1), a type I integral membrane glycoprotein acting as a receptor binding both soluble and immobilized hyaluronan and is involved in lymphatic transport capable of binding to hyaluronic acid (HA), homologous to CD44, the main HA receptor. LYVE1 is a cell surface receptor on lymphatic endothelial cells that can be used as a lymphatic endothelial cell marker. Its expression has also been observed in a subset of macrophages in the meninges of rats [54];

Podoplanin (PDPN), a mucin-type protein with a mass of 36- to 43-kDa. PDPN is relatively well conserved between species, with homologues in humans, mice, rats, dogs and hamsters. PDPN has been found to have functions in lung alveolar cells, kidney podocytes, and lymphatic endothelial cells [55];

Aquaporin-4 (AQP4), a water channel protein belonging to the aquaporin integral membrane protein family, which conduct water through the cell membrane. AQP4 is the most prevalent aquaporin channel, localized in peri-microvessel astrocyte foot processes, glia limitans, and ependymal regions [56].

### 1.3. Myelin Markers

Myelin basic protein (MBP), a protein involved in myelination of the nervous system functioning as an insulator to increase the velocity of axonal impulse conduction. MBP maintains the correct structure of myelin, interacting with the lipids in the myelin membrane [57];

### 1.4. Neuronal Markers

Neurofilament light chain (NF-L), a neuronal cytoplasmic protein used as a marker for amyotrophic lateral sclerosis, multiple sclerosis, Alzheimer’s disease and Huntington’s disease monitoring. NF-L is used as a neuronal biomarker measured in cerebrospinal fluid and reflects axonal damage in a wide variety of neurological disorders [58].

### 1.5. Neurovascular/Blood-Brain-Barrier (BBB) Markers

S100 Calcium Binding Protein-βeta (S100*β*), a protein localized in the cytoplasm and nucleus of a wide range of cells, and is involved in the regulation of a number of cellular processes such as cell cycle progression and differentiation [59];

Platelet-derived growth factor (PDGF), a growth factor regulating cell growth and division in blood vessel formation, growth of blood vessels from already-existing blood vessel tissue, including BBB [60].

### 1.6. Genomic Activation/Repair Markers 

DNA-polymerase-β (POLB), an enzyme that in eukaryotic cells performs base excision repair (BER) required for DNA maintenance, replication, recombination, and drug resistance [61];

RNA-polymerase II (RNAP II), a multiprotein complex that transcribes DNA into precursors of messenger RNA (mRNA) and most small nuclear RNA (snRNA) and microRNA [62].

While we recognize that all the above-defined neurobiological functions are highly interconnected with each other, for the scope of the current investigation, we aimed to explore each neurobiological function separately in order to possibly identify which biological or molecular function could display a higher level of blast-sensitivity, or blast-reactivity, across different regions of the mammalian brain.

In addition, using ELISA quantification methods, we performed measurements of plasma level changes for some of the examined markers (i.e., AQP-4, S100*β*, PDGF, GAP43) to possibly identify peripheral biomarkers directly related to the molecular alterations found in one or more brain regions and as directly generated by two consecutive explosive-driven blast wave exposures.

## 2. Materials and Methods

Detailed information of the study paradigm, time course, blast site, animal transportation, blast set-up apparatus, blast exposure schedule, controlled-blast parameters and procedures, post-blast behavioral protocols and behavioral and neuropathological outcomes are described in Murphy et al. [40]. Here, we briefly describe the general experimental scheme and methods used.

### 2.1. Animals

A total of 24 male Sprague–Dawley rats (Charles River Laboratories, Wilmington, MA, USA) 250–320 grams (2–3 months of age) were used. Animal use protocol was approved by the Institutional Animal Care and Use Committee (IACUC-NEU-15-939) at the Uniformed Services University (USU, Bethesda, MD, USA) in compliance with the PHS Policy on Humane Care and Use of Laboratory Animals, the NIH Guide for the Care and Use of Laboratory Animals, and all applicable federal regulations governing the protection of animals in research.

### 2.2. Study Time Course

Rats arrived approximately 3 weeks prior to the first blast day and divided into 2 groups: double blast-exposed (2×B) (two blast events, 24 h apart, *n* = 12) and control (Ctl) (equivalent handling and anesthesia, but no blast exposure, *n* = 12). During these 3 weeks, baseline behavioral evaluations were conducted for neurological function, cognitive function/spatial learning, locomotor activity and gait analysis [40].

On Day 0, rats were taken to the blast site for the first of 2 blasts or blast-control exposures. Twenty-four (24) hours later, rats were again taken to the blast site for a second exposure. For 2 weeks following the second exposure, rats were assessed on the same battery of tests used prior to the blast. Euthanasia was carried out on Day 15. 

### 2.3. Blast Exposure

At the time of blast, rats in the 2×B group were deeply anesthetized by i.p. administration of a mixture of ketamine (80 mg/kg) and xylazine (10 mg/kg) and placed into a cylindrical aluminum holder allowing exposure to just the head and minimizing the blast impact on other organ systems. The shock wave parameters equated to 30-psi (~207 kPa) incident overpressure with 8–10 msec positive-phase duration, as generated by the detonation of sensitized liquid nitromethane. The explosive charge was placed within the driver section of the blast wave generator at a 20′ distance from the fixed rat holder. Please refer to Murphy et al. for a complete diagram and more detailed description of the blast apparatus [40].

### 2.4. Blood and Tissue Collection

Fifteen days after the second blast, rats were deeply anesthetized by i.p. administration of a mixture of ketamine (80 mg/kg) and xylazine (10 mg/kg). Rats were euthanized via thoracotomy and approximately 4 mL of blood was collected transcardially into EDTA-K2 vacutainer tubes containing 150 μL protease inhibitor cocktail (P2714, Millipore-Sigma, Burlington, MA, USA). The blood samples were centrifuged at 1300× *g* at 22 °C for 10 min. The supernatant containing plasma was removed, transferred to clean tubes and centrifuged once more at 12,000× *g* at 4 °C for 15 min. Plasma was aliquoted and frozen in liquid nitrogen.

Following blood collection, 6 rats from each group (2×B, *n* = 6 and Ctl, *n* = 6) were perfused with 0.9% saline for 5 min to remove blood. Brains were removed, flash frozen and stored at −80 °C for protein analysis. The remaining 6 rats per group (2×B, *n* = 6 and Ctl, *n* = 6) were perfused with 0.9% saline followed by chilled fixative solution (4% paraformaldehyde) for 5 min. Brains were removed, post-fixed and transferred to 20% sucrose for cryoprotection. Fixed brains were flash frozen and stored at −80 °C.

### 2.5. Protein Extraction and Western Blot (WB) Procedures

Fresh frozen brains were cut at 100 µm on a cryostat and four brain regions: frontal cortex (FCtx), hippocampus (H), cerebellum (CRB) and brainstem (BS) were microdissected from these sections [63]. This allowed us to observe the molecular vulnerability across different anatomical regions of the rat brain. Samples were homogenized in Dounce homogenizers with ice cold lysis buffer (1 mL/100 mg tissue; 50 mM Tris–HCl (pH 8), 1% Igepal, 150 mM NaCl, 1 mM EDTA, 1 mM PMSF, 1 mM NaF, 1:100 protease inhibitor cocktail (Sigma-Aldrich, P2714, St. Louis, MO, USA)). Samples were centrifuged at 12,000× *g* for 20 min and supernatants collected, aliquoted, and frozen at −80 °C. Total protein content was determined using the Micro BCA assay (Thermo-Fisher Scientific, 23235, Waltham, MA, USA). An amount of 10–20 µg of protein was loaded on Novex Nupage 4–12% Bis–Tris Gels (Life Technologies, NP0329, Carlsbad, CA, USA) and was run at 200 V constant for 30 min or on Novex Nupage 3–8% Tris Acetate Gels (Life Technologies, EA03785, Carlsbad, CA, USA) at 150 V constant for 60 min (for high molecular weight protein evaluation). Gels were transferred to PVDF membranes using the iBlot2 dry transfer method (Life Technologies, IB21001, Carlsbad, CA, USA). Membranes were blocked in 5% milk in 1 × TBST for 1 h at 25 °C. Primary antibodies (Table 1) were diluted in 5% milk in 1 × TBST and incubated on the membranes overnight at 4 °C. Membranes were rinsed 3 × 5 min in TBST. HRP tagged secondary antibodies (goat anti-mouse [1:2000, Abcam, ab97040, Cambridge, MA, USA], goat anti-rabbit [1:2000, Abcam, ab97080, Cambridge, MA, USA or 1:5000, Proteintech, SA00001-1, Rosemont, IL, USA]) were diluted with 5% milk in 1 × TBST and incubated on the membranes for 1 h at 25 °C. Membranes were rinsed 3 × 5 min in TBST and 1 × 5 min in TBS. Membranes were incubated with chemiluminescent substrate (SuperSignal West Pico Chemiluminescent Substrate, Thermo-Fisher Scientific, 34577, Waltham, MA, USA) for 1 min and imaged on the LiCor C-Digit Blot Scanner (LiCor Biosciences, Lincoln, NE, USA) using the high sensitivity setting (12 min exposure) producing a dynamic range of exposures for each protein of interest. All membranes were stripped 1x with Restore Plus Stripping Buffer (Thermo-Fisher Scientific, 46430, Waltham, MA, USA), for 15 min, rinsed with TBS and processed for immunoblotting as described above using GAPDH (1:40,000, Proteintech, 60004-1, Rosemont, IL, USA) for the loading control. For each marker and for all four brain regions a total of 3 WB runs were performed and averaged. Densitometry was performed with NIH ImageJ software (2.0.0) with all protein signal intensities normalized to GAPDH signal intensity. Signal intensities came from the raw optical density measurements. Table 1 summarizes all antibodies used in the study.

### 2.6. Enzyme Linked Immunosorbent Assays (ELISA)

Following the results from WB assays, we chose to examine in the plasma a panel of some of the same proteins examined in the brain utilizing some of the commercially available rat-specific ELISA kits (Table 2). Plasma was used from the same animals used for WB brain analyses (2×B, *n* = 6 and Ctl, *n* = 6). Each ELISA kit was used following manufacturer’s instructions with all plasma samples run in duplicate at assay appropriate dilutions. The optical density of each assay was read at a 450 nm wavelength on an Infinite M200Pro Spectrophotometer (Tecan, Morrisville, NC, USA). Protein concentrations were determined via regression analysis against the standard curve using “Four Parameter Logistic Curve” online data analysis tool (MyAssays Ltd., 23 February 2022, http://www.myassays.com/four-parameter-logistic-curve.assay). Ultimately, after regression analysis, 2 samples from the 2×B group were determined to be outside the range of the standard curve (to the lower extreme of the curve) and were not included in the final analyses resulting in *n* = 4 for the 2×B group.

### 2.7. Immunofluorescence (IF)

Fixed-frozen brains were cut at 20 µm thickness on a cryostat (Control, *n* = 4, Blast, *n* = 4). Brainstem sections used for immunofluorescence ranged from Bregma −9.72 mm to −11.76 mm [63]. Two brains per group were utilized for testing/optimization and other evaluations leaving four brains per group for these immunofluorescence evaluations. Tissue was first rinsed in 1X PBS 3× 10 min prior to incubation in citrate buffer (pH 6.0) antigen retrieval solution in an oven at 37 °C for 30 min. Slides were removed from oven and continued to incubate on a shaker at room temperature for 20 more minutes. Slides were then rinsed 3× 10 min in 1X PBS before being incubated for one hour in blocking buffer containing 10% normal goat serum (NS) and 0.4% Triton-X. Tissue was incubated with primary antibody for AQP4 (1:100, Abcam, ab128906, Cambridge, MA, USA) overnight at 4 °C. Primary antibody dilutions were made in 1X PBS containing 3% NS. Following incubation, slides were rinsed 3× 10 min in 1X PBS and then incubated for one hour in AlexaFluor secondary antibody at 1:200 dilution, using goat anti-rabbit 594 (AlexaFluor 594, Invitrogen, A11037, Carlsbad, CA, USA) in 1X PBS containing 3% NS. Slides were rinsed 3× 10 min in 1X PBS and coverslipped using Vectashield Vibrance with DAPI. Images were taken on an Olympus microscope using VS120 virtual scanner software (VS-ASW FL v. 2.7, Olympus Corporation, Tokyo, Japan).

### 2.8. Immunofluorescent (IF) Intensity Quantification

AQP4-positive tissues were quantified using the mean fluorescent intensity (MFI) of immunofluorescent images [64]. Using ImageJ (v. 1.53t, National Institutes of Health, NIH, Bethesda, MD, USA), we calculated MFI for whole BS and the dorso-medial BS region, which was defined as being the area subjacent to the 4th ventricle at the midline until the center of the BS. The dorso-medial BS region selected based on IF signal that at a visual fluorescent microscopy inspection appeared to be higher in 2×B vs. Ctl animals. Images were taken at 10× magnification. In order to standardize for any artifact differences that might appear during imaging, one image was chosen and all other images taken were then matched to it for equal intensity. Measurements were taken for two regions: whole BS (coronal area) and 50% of the BS area along the midline (dorsal area), which includes midline nuclei such as raphe obscurus and medial-longitudinal fasciculus (mlf) [63]. For the dorso-medial BS region, the center of the BS was located along the midline and a box of equal width was placed extending from center to the 4th ventricle. The midline was defined as just inferior to the 4th ventricle and included the mlf and dorsal-most tip of the raphe obscurus nuclei. Equal-sized boxes were drawn for each image at the same IF intensity. Each region as well as an unstained background area were traced separately and the mean pixel color was measured. MFI was calculated using the formula: MFI (ROI) − MFI (background) = Final MFI.

We calculated MFI measuring IF intensity for a total of two sections per each animal for a total of four (4) Ctl and four (4) 2×B animals and calculated the average. Additionally, we calculated an index for the smaller ROI (the dorso-medial BS area included in the square) divided by MFI of the whole BS in order to standardize each section to itself. Statistical analysis was performed using GraphPad Prism (v. 9.0.2, GraphPad Software, San Diego, CA, USA).

### 2.9. Statistics

Protein expression levels from both WB, ELISA, and IF intensity were analyzed by 2-tailed, unpaired *t*-tests. Data values reported are mean +/− SEM. Differences with *p*-value ≤ 0.05 were considered significant for all quantification analyses. Statistical tests were performed using GraphPad Prism version 7.03 for Windows (GraphPad Software, La Jolla, CA, USA).

## 3. Results

### 3.1. Molecular Outcomes

#### 3.1.1. Effects of Explosive-Driven Double Blast Exposure on Synaptic-Markers

No significant differences between 2×B and Ctl were identified in protein expression levels for SYN, SPIN, PSD-95 and GAP-43 in all examined regions, except lower levels of SYN were identified in the CRB (*p* = 0.01) and lower levels of PSD95 were identified in the BS (*p* = 0.04) (Figure 1). It is important to notice that two bands were visible for PSD95 in the Western blot and these were evaluated together based on the reported molecular weight range of 95–110 kDa for PSD95. As for GAP43, there was a visible increased trend in the 2×B vs. Ctl in the BS region; however, no statistical significance was reached (*p* = 0.12). Additionally, GAP-43 was the only synaptic marker assessed using ELISA and it was below the level of detection in plasma via ELISA analysis (using plasma concentration of 1:5).

#### 3.1.2. Effects of Explosive-Driven Double Blast Exposure on Glymphatic-Markers

Significant increases of AQP4 in the FCtx (*p* = 0.03) and BS (*p* = 0.0002) and decrease in H (*p* = 0.0001) were identified (Figure 2). Moreover, a detectable, although not significant (*p* = 0.2), increased level AQP4 in plasma was found when comparing 2×B vs. Ctl animals (Figure 3). No significant differences between 2×B and Ctl were identified in protein expression levels for LYVE1 and PDPN across all examined regions.

#### 3.1.3. Effects of Explosive-Driven Double Blast Exposure on Neuronal and Myelination-Markers

No significant differences between 2×B and Ctl were identified in protein expression levels for MBP and NF-L across any of the examined brain regions (Figure 4). Two isoforms of MBP are visible in the western blot, at 21.5 kDa and 18 kDa. These isoforms were evaluated separately and together in the densitometry analyses and no significant differences were found for any of these evaluations.

#### 3.1.4. Effects of Explosive-Driven Double Blast Exposure on Neurovascular Markers

An increased level of S100β (*p* = 0.006) and a decreased level of PDGF (*p* = 0.02) were identified in the BS between 2×B and Ctl. No other differences were noted in any other brain region (Figure 5). S100β and PDGF were both assessed by ELISA and were below the level of detection in plasma via ELISA analysis (using plasma concentration of 1:2 and 1:10 for both proteins).

#### 3.1.5. Effects of Explosive-Driven Double Blast Exposure on Genomic Activation/Repair Markers

POLB was significantly increased in BS (*p* = 0.0086) in 2×B vs. Ctl animals (Figure 6). No differences were identified in protein expression levels for RNAP II between 2×B and Ctl across all examined regions.

### 3.2. Immunofluorescence Analyses

The immunofluorescence (IF) analyses focused on the visualization of AQP4 as expressed across coronal sections of both 2×B and Ctl animals in order to confirm the identified increased level of this marker in the BS and possibly to identify a more specific region of the BS where that increase occurred. Figure 7 shows the IF signal of AQP4 (in red) and its relative higher level of expression in the BS of 2×B in comparison to Ctl animals. In particular, through a visual inspection of the sections at 10× magnification, it appeared that a specific BS sub-region, (the dorso-medial region) demonstrated higher levels of AQP4 IF intensities in 2×B vs. Ctl. Based on this preliminary assessment, we quantified AQP4 IF intensity signals measuring the total BS area (coronal sections) and the dorso-medial region of the BS (See Figure 7). The statistical analyses did not show significant differences of IF intensities between 2×B vs. Ctl animals when the total BS areas were considered. When more specific regions were measured, however, we did see a significant difference in the superior 50% midline (*p* = 0.021) and the medio-dorsal area (*p* = 0.009). Additionally, the comparison between BS dorso-medial area/total BS ratio in the 2×B vs. Ctl animals revealed a significant difference (*p* = 0.048).

## 4. Discussion

These newer and additional findings produced by analyzing the primary effects of two consecutive explosive-driven blasts on the mammalian brain show that, 15 days after the second blast event a series of specific molecular changes in specific regions of the brain particularly sensitive to the primary effects of blast waves such as the BS, are present. We found that neurovascular and DNA-repair markers such as AQP4, S100*β*, PDGF and POLB in comparison to the other markers considered in this study (pre- and post-synaptic, myelin, neuronal markers) were more often altered in the BS of double blasted vs. sham animals in comparison to all other examined brain regions (FCtx, H, and CRB). These BS-associated molecular changes appear to represent direct and possibly long lasting (at least 15 days post second blast exposure in our experiment) abnormal molecular signals induced by exposure to the primary effects of repeated blast waves. Moreover, while we did not plan to identify which specific nucleus or set of nuclei of the BS could be particularly sensitive to the cumulative blast wave primary effects, the IF intensity measurements showed that the dorso-medial region of the BS, including, among others, nuclei such as the raphe obscurus [Ro]—a component of the raphe nuclei—and the medial longitudinal fasciculus (mlf), is particularly vulnerable or reactive to the primary effect of repeated blast waves. The raphe nuclei are usually considered in relationship to serotoninergic alterations in the context of depressive disorders, a notion that would match very well with some of the psychiatric consequences observed in blast-exposed military personnel [65]. It is also important to emphasize that the BS is the neuroanatomical region containing most of the survival-related nuclei and nodes of neurocircuits regulating and maintaining basic physiological functions (i.e., cardio-respiratory centers, chemo-sensors, sleep-awakening cycles, etc.) [66]. Consequently, any pathological process affecting BS nuclei, including the raphe nuclei, mlf and others, can have major repercussions to the rest of the brain, CNS in general, and the rest of the entire organism [67].

As for the observed molecular changes, it is important to highlight that in addition to the BS-specific increased levels of AQP4, S100*β*, and PDGF observed in the double blast animals, the increased level of POLB in the BS is a signal of particular pathogenetic relevance. In fact, higher levels of POLB in the BS are an important confirmatory enzymatic signal demonstrating the particular sensitivity or vulnerability of the BS to the blast wave effects as related to the activation of neural tissue DNA repair mechanisms, which can be triggered by numerous types of brain injury, with the primary effect of blast waves being likely one of them. While POLB is a relatively well-studied molecule and commonly recognized as an essential enzyme activated in complex mechanisms of DNA repair, it is also involved in neurodegenerative mechanisms after injury [68]. To the best of our knowledge, while POLB and DNA-polymerases have been investigated in various impact-TBI studies [69,70], no blast-TBI studies have ever described the expression level changes of either POLB or other DNA-polymerases across different brain regions in response to repeated explosive-driven primary blast wave exposure.

As for the S100*β* changes, apart from its well-established role in neurovascular processes [71,72], the higher levels of S100*β* detected in 2×B animals could also be associated with synaptogenesis phenomena after TBI [73]. It is important to note that S100*β* changes related to TBI events have been described mainly as plasma biomarker changes in different impact- and blast-TBI studies in humans and animals [74,75,76]. By contrast, S100*β* expression level changes measured directly using brain tissue from different brain regions of animals exposed to repeated explosive-driven primary blast waves have been much rarer, if any. In fact, in an attempt to compare our findings with other blast-TBI studies in rats or other experimental animals, we observed that currently, no investigations have ever reported data on S100*β* levels as directly measured in different brain regions using repeated explosive-driven primary blast wave experimental paradigms. We retain that this last new aspect is of particular importance since multiple studies showed how the timing between blast event and blood collection and assays used, TBI modalities, tissue-organ biomarker source specificity, aging, genetic, pre-existing and other possible environmental factors, can all impact plasma level measurements of S100*β* (as well as of other biomarkers) and so posing some concerns on the reliability for some of the brain pathology-plasma correlations currently accepted as diagnostic or prognostic tools for specific post-TBI events occurring in the brain [77]. By contrast, our findings represent a rare set of data from neuroanatomically-based dissections associated with possible direct S100*β* plasma level correlates. However, despite altered levels of S100*β* observed in the corresponding 2×B brains and BS region in particular, S100*β* plasma levels (by ELISA method) were not detectable in our samples. This latter finding underscores the necessity to perform more precise brain-plasma correlation analyses in blast-TBI studies to provide a better mechanistic understanding and data reliability for each specific plasma biomarker identified as related to a specific brain region or set of brain regions as the main source of blast-induced brain pathology-plasma changes.

Our findings, which describe neuroanatomical-based molecular alterations directly and distinctively induced by the primary effects of double explosive-driven blast waves, are consistent with some previous data describing increased plasma levels for some of the proteins considered in this study, in humans and animals, after blast-TBI and impact-TBI exposure [78,79,80]. As further novelty here, we report that plasma changes of specific molecules (i.e., AQP4) directly generated by repeated blast waves might originate mainly, although not exclusively, from a restricted set of brain regions, in particular the BS region. However, it would be important to emphasize that the source of AQP4 could be represented not only by neuronal origin but also by other processes such as exfoliated immune or endothelial cells.

In our study, ELISA analyses showed a trend, although not significant, for higher plasma levels of AQP4 that were associated with higher levels of AQP4 in the BS. If our experimental findings can be confirmed by larger studies analyzing brain and plasma samples from humans, they will represent the rationale for a possible and reliable model of brain pathology-plasma correlation investigations. These studies could also represent a valuable method for in vivo pathologic-biochemical assessments, which could directly connect intracranial molecular blast-related changes in specific brain regions with plasma level alterations. In addition, these direct double blast-induced BS-plasma AQP4 expression level changes could have an immediate utility in clinical research and even clinical settings after specificity and sensitivity validation tests will be performed in blast-TBI human studies [43]. Furthermore, as related to the particular blast sensitivity of the BS, our data seem to better explain the difficulties into identifying specific blast-related lesions using neuroimaging methods, especially MRI, which are notoriously limited to adequately assess the BS region [81].

A wider generalization of our new results would propose that specific regions of the mammalian brain are more vulnerable than others in response to repeated blast wave exposure. In our experimental paradigm, the BS appears to be the brain region more often involved in reacting to the repeated blast exposure vs. other brain regions such as H or CRB, which also show significant alterations. Importantly, it remains unclear if these molecular changes in the BS and all other regions actually characterize reactive, reparative, or compensatory phenomena to the primary effects of blast waves. In this context, it is also important to consider that factors such as the geometry and fluid dynamics of the blast waves could theoretically produce different molecular outcomes in different brain regions based on the original direction and dynamics of the blast wave generated by specific types of detonation [82].

We speculate that higher levels of AQP4 in the BS are signals of compensatory mechanisms in the context of a substantial synaptic stability and functionality as suggested by the lack of meaningful expression level changes of pre- and post-synaptic markers across most of the examined regions, with very few exceptions (see Results). These findings acquire relevance since AQP4 has already been shown to be involved in reparative, regenerative, and neuroplasticity processes [83,84].

Additionally, it has been shown that AQP4 is a molecule also involved in astrocytic and glymphatic drainage mechanisms, especially in tau protein drainage/clearance mechanisms [85]. Intriguingly, tau was one of the proteins previously found significantly altered in double blast vs. control animals [40]. Moreover, while levels of AQP4 tend to diminish with age and are associated with a reduction of clearance capacities [86], in our experiment the absence of age difference between the two experimental groups favors an actual increased expression of AQP4 levels rather than a reduced clearance mechanism in response to increased levels of different types of proteins, and in particular tau [87]. Other blast-TBI studies reported significant changes in AQP4 brain levels as well [88,89]. Although blast intensities, blast waves modalities, and anatomical regions analyzed were different across all those studies, it seems clear that AQP4 changes represent a constant and common molecular change in blast-TBI outcomes and that the glymphatic and blood brain barrier systems can be severely affected by primary blast waves, specifically in more sensitive brain regions such as the BS, which for its anatomical position and structure is particularly predisposed to the shearing effect due to the wave impact.

Importantly, it is not known yet, if in humans AQP4-related alterations generated by multiple blast exposures have a long-term effect on cognitive and behavioral skills [90,91,92]. Future longitudinal pathologic-behavioral studies are necessary to confirm the occurrence of long-term cognitive and behavioral effects as well as their timings and molecular correlates directly generated by repeated explosive-driven blast wave exposure. Recently, a human study using MRI methods showed that the glymphatic system, which usually AQP4 is considered to belong to, does mediate cognitive dysfunction in Alzheimer’s disease [93].

As for PDGF, whose alterations are considered molecular signals associated with vascular reactivity and blood-brain-barrier permeability processes [94,95], we retain that the higher levels of PDGF, together with higher levels of S100*β*, detected in the double blast animals fit well with a series of observations from human studies showing that one of the effects of blast wave exposure is the induction of blood-arterial spasms likely caused by post-blast ischemic events in different brain regions [96,97]. Our data and other neurovascular findings obtained in human samples, seem to converge towards a general pathophysiological hypothesis, which postulates an early and preeminent effect of the blast waves on the blood-brain-barrier (BBB) and neurovascular system [98,99]. Furthermore, the observed increased levels of both S100*β* and PDGF in the BS fit well also with the fact that both molecules are involved in astrocytic-related functions, and astrocytic blood vessel unsheathing processes in particular [100,101]. Specifically, as for the PDGF, which is significantly reduced during processes linked to AD [95], it has been shown that it is also involved in BBB protective molecular mechanisms after a stroke [102]. These last findings emphasize that not only the BS is possibly one of the most, if not the most, blast-sensitive region of the brain, but that multiple reacting neurovascular mechanisms are in place during the relatively acute post-blast exposure period. It is further important to emphasize that, to the best of our knowledge, this is the first time that PDGF expression level changes have been reported as measurements performed directly in specific brain regions from animals exposed to repeated explosive-driven primary blast waves. A recent study using repeated blast exposure produced by a shockwave generator system was not able to detect changes of PDGF-B in rat brains [103]. However, while the experimental parameters between Uzunalli et al.’s study and ours were different, the discrepancy between the PDGF results could be due to the specific modality of the blast generation, emphasizing once again the importance of using more realistic blast wave generators such as the explosive-driven systems for blast-TBI analyses.

As for the GAP-43 changes, which often correlate with NF-L changes in the context of neurodegenerative processes [104], our results seem to corroborate this type of correlation since we did not observe any dissociation between GAP-43 and NF-L changes across any region. Interestingly, though, some recent data showed that a differential expression between GAP-43 and NF-L is actually possible [105] as related to the role of GAP-43 in neuroplasticity (synaptic) mechanisms and so not necessarily related to axonal damage-related processes [53,106,107]. This last observation is in line with previous observations, mainly from impact-TBI experiments, showing that synaptic alterations (i.e., GAP43) across the brain are indeed present and of relevance in the context of impact-TBI [108,109]. By contrast, data on synaptic markers in animals repeatedly exposed to primary blast waves have rarely been reported, with the exception of GAP-43 and other synaptic markers as part of gene arrays in secondary analyses of single blast events generated by a shock tube [110].

## 5. Conclusions

These new data illustrate the presence of cerebrovascular-related compensatory or reparative mechanisms characterized by brain-region specific molecular changes resulting from repeated primary blast wave exposures and the occurrence of an asymptomatic blast-induced molecular altered status (ABIMAS). It remains to be determined though, how long ABIMAS would last and if its exacerbation caused by additional unfavorable events (e.g., further blast and non-blast TBI, genetic or environmental risk factors) might contribute to the later manifestation of neurological and/or psychiatric disorders as described in multiple blast-TBI exposed subjects. Our analyses also show that the blast-sensitivity can vary across different regions of the brain, among which, the brainstem appears to be one of the most, if not the most, blast-sensitive region of the brain.

## Figures and Tables

**Figure 1 brainsci-13-00286-f001:**
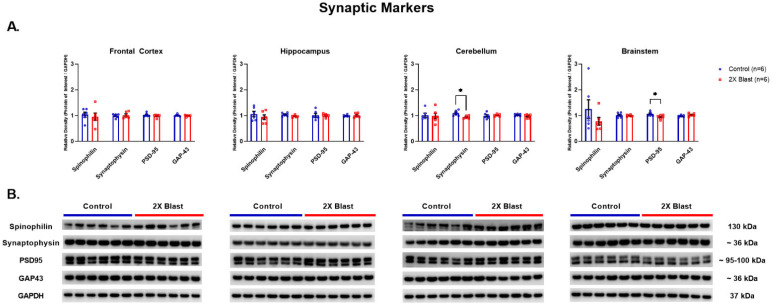
Synaptic related protein expression changes. (**A**) Densitometric ratio of the levels of Spinophilin (single band at 130 kDa), Synaptophysin, PSD-95 (both visible bands evaluated together) and GAP-43 with respect to GAPDH as measured in FCtx, H, CRB and BS 15 days following an explosive-driven double blast exposure, *n* = 6 per group, * indicates *p* values < 0.05 as determined by 2-tailed, unpaired *t*-tests. Error bars represent standard error of the mean (SEM). (**B**) Representative western blots ^#^ for each antibody used. All gels were run in triplicate and data represents the average of 3 runs per sample. ^#^ For full length blots, see Appendix A.

**Figure 2 brainsci-13-00286-f002:**
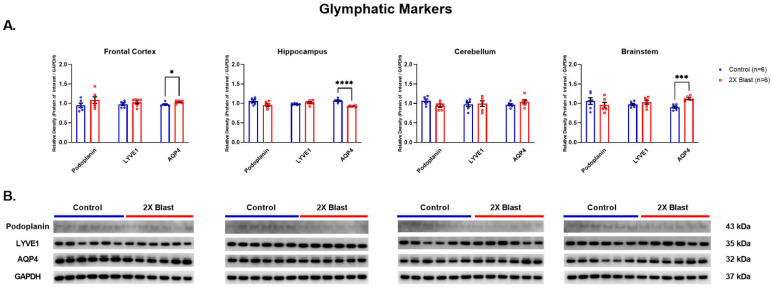
Glymphatic related protein expression changes. (**A**) Densitometric ratio of the levels of Podoplanin, LYVE1, and AQP4 with respect to GAPDH as measured in the FCtx, H, CRB and BS 15 days following an explosive-driven double blast exposure, *n* = 6 per group, * indicates *p* values < 0.05, *** indicates *p* values < 0.001 and **** indicates *p* values < 0.0001, as determined by 2-tailed, unpaired *t*-tests. Error bars represent standard error of the mean (SEM). (**B**) Representative western blots ^#^ for each antibody used. All gels were run in triplicate and data represents the average of 3 runs per sample. ^#^ For full length blots, see Appendix A.

**Figure 3 brainsci-13-00286-f003:**
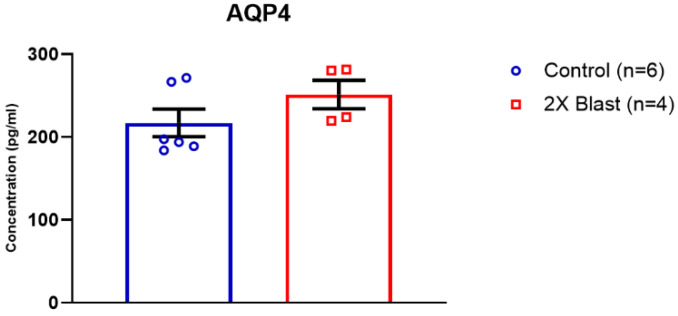
Plasma concentration of AQP4 following double blast exposure. Histogram representing the plasma concentration of AQP4 15 days following an explosive-driven double blast exposure, Ctl, *n* = 6, Blast, *n* = 4.

**Figure 4 brainsci-13-00286-f004:**
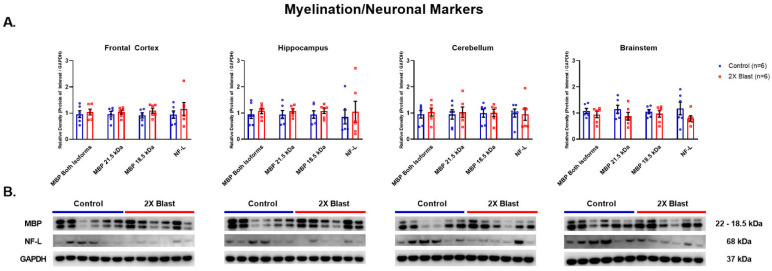
Myelin and neuronal related protein expression changes. (**A**) Densitometric ratio of the levels of MBP, isoforms 21.5 kDa and 18 kDa, (isoforms were evaluated together and separately) and NF-L with respect to GAPDH as measured in the FCtx, H, CRB and BS 15 days following an explosive-driven double blast exposure, *n* = 6 per group. Error bars represent standard error of the mean (SEM). (**B**) Representative western blots ^#^ for each antibody used. All gels were run in triplicate and data represents the average of 3 runs per sample. ^#^ For full length blots, see Appendix A.

**Figure 5 brainsci-13-00286-f005:**
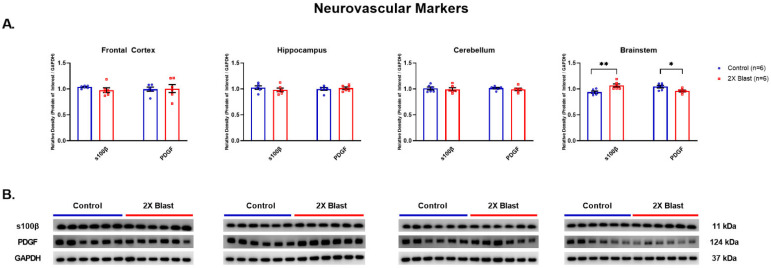
Neurovascular related protein expression changes. (**A**) Densitometric ratio of the levels of S100β and PDGF with respect to GAPDH as measured in the FCtx, H, CRB and BS 15 days following an explosive-driven double blast exposure, *n* = 6 per group, * indicates *p* values < 0.05 and ** indicates *p* values < 0.01 as determined by 2-tailed, unpaired *t*-tests. Error bars represent standard error of the mean (SEM). (**B**) Representative western blots ^#^ for each antibody used. All gels were run in triplicate and data represents the average of 3 runs per sample. ^#^ For full length blots, see Appendix A.

**Figure 6 brainsci-13-00286-f006:**
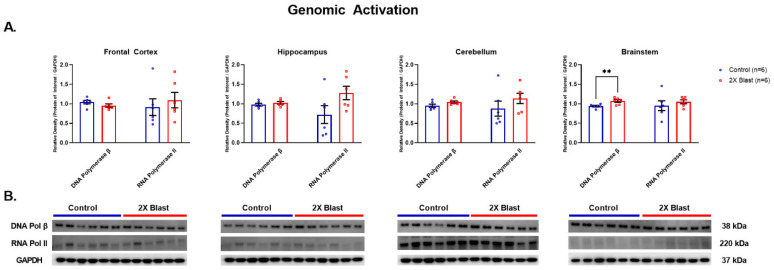
Genomic activation/repair related protein expression changes. (**A**) Densitometric ratio of the levels of DNA Polymerase β and RNA Polymerase II with respect to GAPDH as measured in the FCtx, H, CRB and BS 15-days following an explosive-driven double blast exposure, *n* = 6 per group, ** indicates *p* values < 0.01 as determined by 2-tailed, unpaired *t*-tests. Error bars represent standard error of the mean (SEM). (**B**) Representative western blots ^#^ for each antibody used. All gels were run in triplicate and data represents the average of 3 runs per sample. ^#^ For full length blots, see Appendix A.

**Figure 7 brainsci-13-00286-f007:**
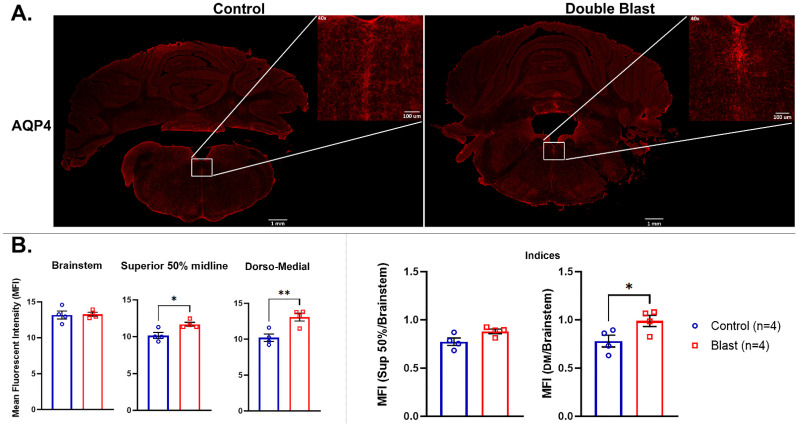
AQP4 IF intensity changes in the BS following double blast exposure. AQP4 in control (Ctl.) versus double blast (2×B) rat BS. (**A**) AQP4-ir positive staining in the BS of a control rat not exposed to blast versus BS of a rat exposed to double blast, taken at 10× magnification. The insets at 40× magnification, defined as superior raphe obscurus-mfl, indicate the area (dorso-medial area) of MFI quantification. (**B**) Graphs showing MFI of total BS, 50% superior BS midline, and dorso-medial area including the superior portion of the raphe obscurus nuclei and mlf. * indicates *p* values < 0.05 and ** indicates *p* values < 0.01 as determined by 2-tailed, unpaired *t*-tests. Indices are MFI of each ROI over MFI of the total brainstem. IF = Immunofluorescence; BS = brainstem; mlf = medial longitudinal fasciculus; indices = IF intensity ratios between total BS/dorso-medial areas; MFI = mean fluorescent intensity.

**Table 1 brainsci-13-00286-t001:** Antibodies and related information for Western Blots (WB).

	Antibody	Protein Concentration Loaded	Antibody Concentration	Cat. #	Source
Synaptic	Spinophilin	10 μg	1:1000	14136	Cell Signaling, Danvers, MA, USA
	Synaptophysin	10 μg	1:2000	ab8049	Abcam, Cambridge, MA, USA
	PSD-95	10 μg	1:1000	75-028	Antibodies, Inc., Davis, CA, USA
	GAP-43	10 μg	1:40,000	ab75810	Abcam, Cambridge, MA, USA
Glymphatic	Podoplanin	20 μg	1:1000	131216	Abcam, Cambridge, MA, USA
	LYVE1	10 μg	1:1000	ab183501	Abcam, Cambridge, MA, USA
	Aquaporin-4 (B5)	10 μg	1:500	sc-390488	Santa Cruz Biotechnology, Dallas, TX, USA
Axonal/Myelin	MBP	10 μg	1:4000	ab62631	Abcam, Cambridge, MA, USA
	Neurofilament Light	20 μg	1:1000	MCA-DA2	Encor, Gainesville, FL, USA
Neurovascular	S100β	10 μg	1:5000	ab52642	Abcam, Cambridge, MA, USA
	PDGF	10 μg	1:5000	ab32570	Abcam, Cambridge, MA, USA
Genomic	DNA Polymerase β	10 μg	1:1000	ab26343	Abcam, Cambridge, MA, USA
	RNA Polymerase II	10 μg	1:500	05-623Z	Millipore-Sigma, Burlington, MA, USA

Type, concentration, and commercial information for all antibodies used in the study.

**Table 2 brainsci-13-00286-t002:** ELISA kits.

Antibody	Cat. #	Source
GAP-43	EKL61628	Biomatik USA, Wilmington, DE, USA
AQP4	abx061777	Abbexa, Houston, TX, USA
S100β	abx256298	Abbexa, Houston, TX, USA
PDGF	KE10057	Proteintech, Rosemont, IL, USA

Type and commercial information for all ELISA kits used in the study.

## Data Availability

The datasets used and/or analyzed during the current study and supporting the conclusions of this article are included in this article and in all Appendix A provided. These datasets are also available from the corresponding author on reasonable request.

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
