# Peer review of "Double Blast Wave Primary Effect on Synaptic, Glymphatic, Myelin, Neuronal and Neurovascular Markers"

_brainsci, 2023, doi:10.3390/brainsci13020286_

Round 1

Reviewer 1 Report

This is a well written manuscript that reports results of the analysis of synaptic, glymphatic, myelin, neuronal and neurovascular markers in rats subjected to double blast waves.  The study is well designed, and the results are clearly presented.

This is a well written manuscript that aims to study the neuropathologic and molecular consequences of blast waves TBI generated using an explosive-driven blast-generating system. A better understanding of the molecular and pathological changes driven by blast induced TBI is an important area of research that has implications for both the military and civilian population alike. By using an experimental setting that employs an explosive-driven repeated blast paradigm that more closely resembles battlefield-like experimental paradigm this report is uniquely positioned to more precisely investigate the direct neuropathological and neuromolecular consequences generated by the primary effects of explosive-driven double blast wave exposure.
The methodology is appropriate and proper controls are used to support the authors conclusion that blast-generated changes are preferentially associated with neurovascular, glymphatic, and DNA repair markers, especially in the Brainstem.
References are properly cited. Data is clearly presented in the figures.

Author Response

We thank this reviewer for his/her extremely positive feedback and for having appreciated the novelty of the data and our efforts to conduct this type of experiment and analyses. 

Reviewer 2 Report

The authors have investigated the expression of synaptic, glymphatic, myelin, neuronal, and neurovascular markers after a double blast model of traumatic brain injury in rats. The manuscript is quite detailed, perhaps overly so. The findings are important, as blast injury is a significant cause of traumatic brain injury in warfare and civilian life. 

The manuscript could be improved with several changes, as follows:

1. In section 3.1.2, lines 404-405, the authors report a trend of increase of LYVE1 in the brainstem. However, a p value of 0.07 does not indicate "a trend" when a p of < 0.05 is accepted as the level of significance in an analysis; it indicates that the value is not significantly different than what would be expected by chance. The text should be modified to reflect appropriate use of statistics.

2. The DISCUSSION is a bit lengthy. The manuscript would be more readable with some editing for length.

3. The CONCLUSION is too long. No statements requiring reference should be included, and supposition should not be included. The CONCLUSION should reflect a brief summary of what the analysis showed. 

Author Response

We thank this reviewer for his/her positive comments and observations.

We agree with this reviewer and we have deleted now the phrase referring to "the trend" for  LYVE1 based on p=0.07. See 404-405 tracked changes.

We appreciated the suggestions about the length of the Discussion and Conclusion, which have been now both reduced. 

Reviewer 3 Report

The objective of the study was to evaluate blast wave effects of experimental blast exposure on neuromolecular profile of brain-related molecules across the different regions of the rat brains. The main advantages of the paper includes the observation of a broad brain area for sensitivity to various molecular  changes in response to repeated blast wave exposure vs sham rats. The data confirm the clinically-related hypothesis by the authors that the blast wave may generate molecular biomarker profiles in four brain regions that may mimic subclinical changes in people exposed to a blast. The meritoriousness of the paper is the discovering quantitative alterations of various glymphatic, synaptic, neuronal, myelin, neurovascular and genome repair markers specific for different brain areas and blood including  AQP4 and other molecules using several antibody-assisted methods of protein identification. The drawback of the paper includes limited discussion on AQP4 origination in plasma that has been skewed to the brain origin only although the molecule may be exfoliated from immune or endothelial cells.

Author Response

We thank this reviewer for his/her positive feedback and valuable comments. 

We agree about the possibility that AQP4 source in plasma can be also from non-neural tissue origin. We have now added a phrase about this important point. Please, see line 571-573.